# Doppler Tomography of the Circumstellar Disk of the Be Star κ Draconis

**Ilfa A. Gabitova** [1], **Anatoly S. Miroshnichenko** [2,3], **Sergey V. Zharikov** [4], **Ainash Amantayeva** [1,*] and **Serik A. Khokhlov** [1]

1 Faculty of Physics and Technology, Al-Farabi Kazakh National University, Al-Farabi Ave., 71, Almaty 050040, Kazakhstan; ilfa3110@gmail.com (I.A.G.); skhokh88@gmail.com (S.A.K.)
2 Department of Physics and Astronomy, University of North Carolina—Greensboro, Greensboro, NC 27402, USA; a_mirosh@uncg.edu
3 Fesenkov Astrophysical Institute, Observatory, 23, Almaty 050020, Kazakhstan
4 Observatorio Astronomico Nacional, Instituto de Astronomia, Universidad Nacional Autonoma de Mexico, Ensenada 22800, BC, Mexico; zhar@astrosen.unam.mx
* Correspondence: amantayevainash@gmail.com

**Abstract:** κ Draconis is a binary system with a classical Be star as the primary component. Its emission-line spectrum consists of hydrogen lines, notably the Hα line with peak intensity ratio (V/R) variations phase-locked with the orbital period P = 61.55 days. Among binaries demonstrating the Be phenomenon, κ Dra stands out as one of a few systems with a discernible mass of its secondary component. Based on more than 200 spectra obtained in 2014–2023, we verified the physical parameters and constructed the mass function. We used part of these data obtained in 2014–2021 to investigate regions in the circumstellar disk of the primary component that emit the Hα line using the Doppler tomography method. The results show that the disk has a non-uniform density distribution with a prominent enhancement at $V_y \approx 99 \, \text{km s}^{-1}$ and $V_x \approx -6 \, \text{km s}^{-1}$ that corresponds to a cloud-like source of the double-peaked Hα line profile. We argue that this enhancement's motion is responsible for the periodic variations in the Hα V/R ratio, which is synchronised in orbital phase with the radial velocity (RV) of absorption lines from the atmosphere of the primary component.

**Keywords:** spectroscopy; Doppler tomography; binary system; emission-line stars; circumstellar matter; variable stars

## 1. Introduction

Classical Be stars are B-type non-supergiants distinguished by their rapid rotation and emission lines in the spectra, along with a notable infrared excess [1]. The pioneering model proposed by Struve [2] attributes these emission lines to recombination processes within a gaseous, geometrically thin, circumstellar equatorial disk. The hypothesis that Be stars could be members of binary systems that have undergone mass transfer emerged as an explanation for their rapid rotation [3]. It suggests that the mass and angular momentum transfer from an originally more massive secondary component could cause the primary component to spin up and eventually become a Be star. It is reasonable to expect that the remaining stripped core of the companion, classified as a subdwarf star, can influence the spectral characteristics of Be stars through tidal effects. The latter may also cause the truncation of the circumstellar disk around the Be star [4].

Studies on the circumstellar disks of Be stars are as important as the exploration of their binarity, because the two subjects are connected through a tidal interaction between the system's components. While it is well-established that the disks serve as sources of line emission, including the Hα line, critical gaps remain in our understanding of the intensity distribution, structural configuration, and essential dynamical characteristics. Many Be stars demonstrate periodic variations in the intensity ratio of the Hα emission

peaks, commonly referred to as the V/R ratio, where "V" stands for the "violet" peak, and "R" stands for the "red" peak.

One way to explain these variations is the rotation of a fixed non-axisymmetric density structure in the circumstellar disk that surrounds the Be star. Such a structure can take the form of a one- [5] or two-armed density spiral [6,7] or a hot spot [8]. Disks in Be stars exhibit another feature: in a noticeable number of such systems, they tend to disappear for several years and reappear again suddenly. The mechanisms governing this phenomenon remain uncertain. A leading theory involves oscillations of the one-armed spiral structure in a Keplerian disk proposed by Okazaki [5], but the causes of these oscillations are still unknown. Other possible explanations for long-term V/R variations are presented by Telting [9].

The object of this study, $\kappa$ Draconis ($\kappa$ Dra, HD 109387, HR 4787, 5 Dra), is a classical Be star with a long history of observations that started as early as 1888, according to Jessup [10], who concluded that variations in the intensity of emission lines (see Figure 1) are cyclic with a period of $\approx$23 years. This behaviour was confirmed by Juza et al. [11] and, later, Saad et al. [12] corrected the cycle length to $\approx$22 years.

In 1991, Juza et al. [13] first reported $\kappa$ Dra as a binary system with a circular orbit and an orbital period of P = 61.55 days. Saad et al. [14] found that V/R variations in the H$\alpha$ line are phase-locked with the orbital period. This result was recently confirmed by Miroshnichenko et al. [15]. The most accurate fundamental parameters of the system were provided by Klement et al. [16].

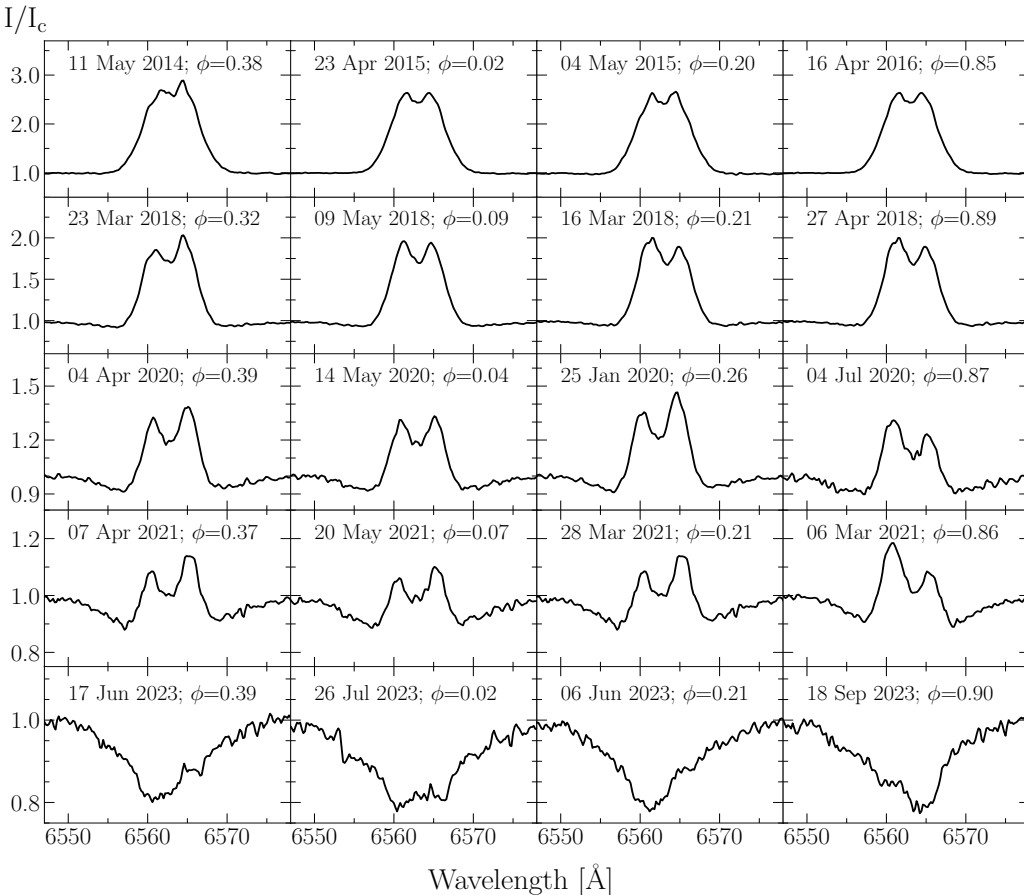

**Figure 1.** Examples of the H$\alpha$ line profile in the spectra of $\kappa$ Dra. The observational dates are shown in each panel. For more on the choice of the zero-phase epoch, see Section 3.

In this paper, we explore the circumstellar disk structure of the $\kappa$ Dra binary system using the Doppler tomography method [17]. In Section 2, we describe our observations and

the process of data reduction. In Section 3, we describe our analysis of the periodic V/R variations and radial velocities (RV) of the Hα line as well as of the RV variations in a set of absorption lines. The circumstellar disk study is also presented in Section 3. The results are further discussed in Section 4. Finally, our conclusions are presented in Section 5.

## 2. Observations and Data Reduction

The study is based on the spectroscopic data obtained using an échelle spectrograph attached to the 0.81 ṁ telescope of the Three College Observatory (TCO) located in the central part of North Carolina, USA. Detailed information about TCO's equipment and observational program can be found in [18]. The spectra were taken between 2014 and 2023 with a spectral resolving power of $R \sim 12,000$. The spectral range spans from $\sim 3740/4250$ Å to $\sim 7890$ Å without gaps between spectral orders. A typical amount of time for an individual exposure was 180–300 s, and each spectrum consisted of several such exposures. The processing of the spectra was carried out using the *echelle* task in IRAF. The wavelength calibration was performed using a ThAr lamp.

A large fraction of our spectra in the Hα region are affected by telluric lines, particularly those acquired during late spring to early autumn seasons. To perform the V/R intensity ratio measurements, we originally divided our spectra using templates of telluric lines for different humidity levels created through the interpolation of Gaussians corresponding to each telluric line. However, despite $\sim 70\%$ of the Hα line profiles being contaminated in the peaks area, the results of the V/R measurements remained largely unaffected compared to the analysis of the same spectra before the cleaning procedure.

As a result, a total of 223 TCO spectra were analysed. Of these, only 101 spectra were employed for the Hα V/R measurements as well as for the Doppler tomography due to the disappearance of the line emission in 2023. We also used 30 spectra taken in 2014–2016 and retrieved from the BeSS (Be Star Spectra) database[1] with $R \sim 9000$–16,000 to fill some gaps in our data. All the spectra were interpolated with an increment of $\Delta\lambda = 0.2$ Å.

## 3. Hα Line Evolution and Doppler Tomography

In Figure 1, we present the selected Hα line profiles taken at different epochs and in different orbital phases. The rows of the plot show profiles with about the same intensity, while the columns from top to bottom follow the decreasing disk contribution with time in similar phases. It is clearly visible that the line intensity decreases with time, while variations in the V/R ratio remain phase-locked with the orbital period. The changes in the Hα line flux over the period of our observations are illustrated by the evolution of its equivalent width (EW) in the bottom panel of Figure 2. However, the V/R ratio shows a stable sine-like behavior in relation to the orbital phase even when the emission component is barely above the continuum level (Figure 2, top panel). A similar graph of the Hα EW changes was published in Klement et al. [16] from a different dataset but with no analysis of the V/R variations.

From the RV curve of absorption lines in the spectral region $4370-4500$ Å, we derived the orbital period P = 61.55(4) days and the semi-amplitude $K_1 = 6.33(25)$ km s$^{-1}$ for the primary component (see [15] for details). The latter parameters give the mass function f(M$_1$) = 0.0016(2) M$_\odot$ close to those recently derived by Saad et al. [19] f(M$_1$) = 0.0021(1) M$_\odot$ and Klement et al. [16] f(M$_1$) = 0.0020(2) M$_\odot$. Thus, in our analysis, we used the components' masses M$_1$ = 3.65(48) M$_\odot$ and M$_2$ = 0.43(4) M$_\odot$, as well as the orbital inclination angle $i = 50°0(3°4)$ from Klement et al. [16].

It is thought that the source of the Hα emission line in Be stars is a gaseous self-ejected (decretion) circumstellar disk. A trailed spectrum of a spectral region around the Hα line folded with the orbital period P = 61.55 days clearly shows a double-peaked structure of the line profile and the presence of an S-wave, which is typically associated with a hot spot in the disk (Figure 3, middle panels; also, see [20] for examples). By assuming a Keplerian velocity field, we can project our phase-resolved spectroscopy onto a predefined velocity framework to construct the Doppler tomography of the system, as proposed by Marsh and

Horne [17]. Traditional objects of Doppler tomography are accretion disks in cataclysmic variables. Nevertheless, despite the fact that Be stars in binary systems usually have large separations between the components and have likely already undergone the process of mass transfer, Doppler tomography is still proven to be helpful for studying circumstellar disks as a source of the Hα emission line [21].

**Table 1.** Orbital parameters of the κ Dra binary system .

| Parameter | Value |
|---|---|
| P [days] | $61.55 \pm 0.04$ |
| T0 [HJD] | $2{,}459{,}074.45 \pm 0.43$ |
| $\gamma$ [km s$^{-1}$] | $-4.86 \pm 0.18$ |
| $K_1$ [km s$^{-1}$] | $6.33 \pm 0.25$ |
| $f(M_1)$ [M$_\odot$] | $0.0016 \pm 0.0002$ |

Parameters listed are as follows: P—orbital period, T0—epoch of the inferior conjunction of the primary component, $\gamma$—systemic velocity, $K_1$—semi-amplitude of the RV variations of the primary component, $f(M_1)$—mass function. These parameters are derived by our team [15] and based on the RVs of absorption lines in the 4370–4500 Å region measured by the cross-correlation method using the task *rvsao* in IRAF.

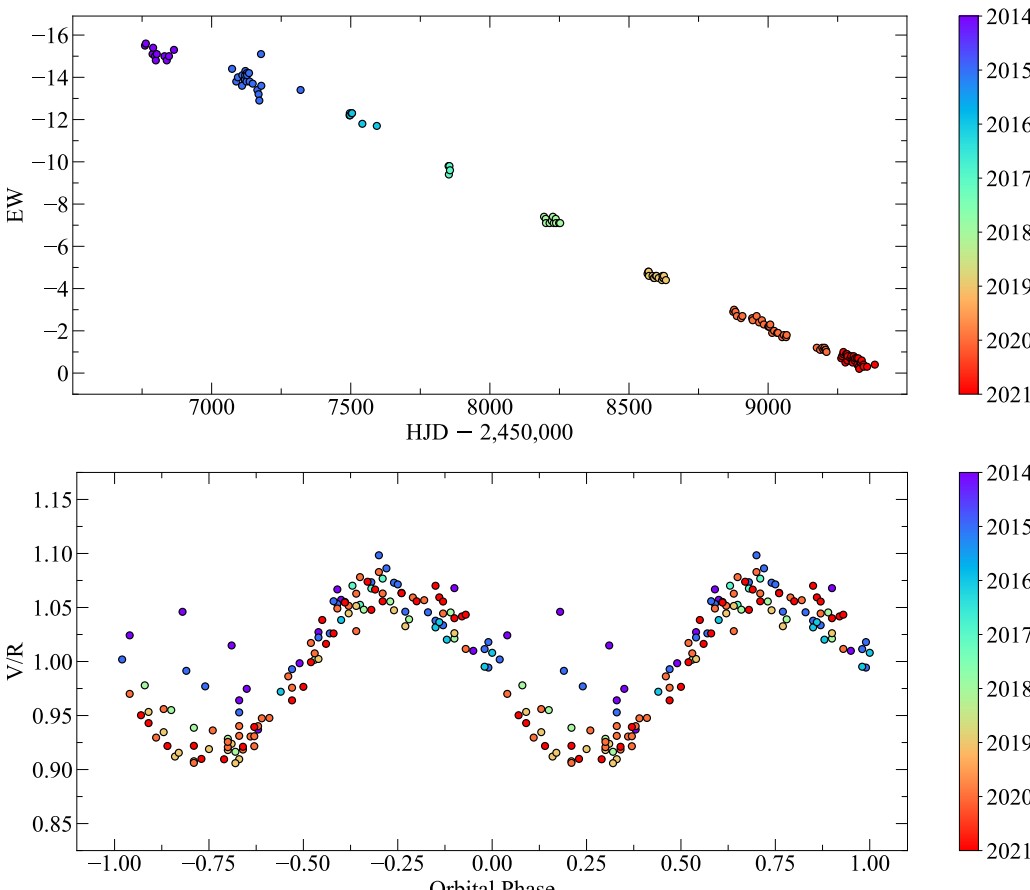

**Figure 2.** (**Top**): V/R variations folded with the orbital period 61.55 days. (**Bottom**): Temporal variations in the equivalent width (EW) of the Hα line. The colors correspond to observation dates, as shown on the bar on the right.

The Doppler tomography of the κ Dra system was constructed using the system parameters from Klement et al. [16] and those listed in Table 1. Our data provide a dense coverage of orbital phases and span over 40 orbital cycles. The procedure was executed via the *dopmap* program developed by Spruit [22]. In total, 131 spectra (see Section 2) from

several time intervals between 2014 and 2021 were used for this purpose. The emission line became too weak for measurements after 2021.

**Table 2.** Parameters of the hot spot and corresponding disk radii in different years.

| Observation Periods | $V_x$ [km s$^{-1}$] | $V_y$ [km s$^{-1}$] | $V$ [km s$^{-1}$] | $V_{spot}$ [km s$^{-1}$] | $R_d$ [$R_\odot$] |
|---|---|---|---|---|---|
| 2014–2017 | $-4 \pm 52$ | $80 \pm 52$ | $80 \pm 74$ | $104 \pm 97$ | 64 |
| 2017–2019 | $-13 \pm 35$ | $100 \pm 28$ | $101 \pm 45$ | $132 \pm 59$ | 40 |
| 2019–2021 | $1 \pm 36$ | $117 \pm 28$ | $117 \pm 46$ | $153 \pm 60$ | 30 |

The following hot spot parameters are listed: $V_x$ and $V_y$—coordinates of the hot spot on the Doppler maps; $V=\sqrt{V_x^2 + V_y^2}$—its velocities; $V_{spot} = V/\sin i$—the hot spot velocity corrected for the system inclination; and $R_d$—the disk radius calculated using Equation (2). The inclination angle of the system is adopted as $i = 50°.0$.

We divided our spectral data into three slightly overlapping blocks to explore the disk structure evolution. The first block (2014–2017) corresponds to $EW_{H_\alpha} \lesssim -9$ Å, the second one (2017–2019) covers a range of $EW_{H_\alpha}$ from $\approx -10$ Å to $\approx -4$ Å, and the last one (2019−2021) includes cases where $EW_{H_\alpha} \gtrsim -5$ Å. To exclude the absorption part of the line, we subtracted a model spectrum of the Hα line from all spectra (see Figure 3, left panels). The model spectrum of the stellar atmosphere was constructed via the SPECTRUM program [23] using Kurucz/Castelli data [24] corresponding to a star with the following parameters: $T_{eff}$ = 14,000 K, log g = 3.5, and $v \sin i$ = 200 km s$^{-1}$. These parameters are close to those provided by Klement et al. [16].

In the left panel of Figure 3, we show the average resulting line profiles (red lines). In the right panels of Figure 3, we present Doppler maps obtained from 44 spectra taken in 2014–2017 (top), 31 spectra taken in 2017–2019 (middle), and 76 spectra taken in 2019–2021 (bottom). As seen from the trailed spectra, orbital phases are well covered. All the Doppler maps show the presence of a torus-like structure and an extended bright spot at $V_y \approx 100$ km s$^{-1}$ and $V_x \approx 0$ km s$^{-1}$. The map for 2014−2017 shows a non-uniform ring of the disk emission ($V \approx 100$ km s$^{-1}$) with a prominent intensity enhancement centered at $V_y \approx 80$ km s$^{-1}$ and $V_x \approx -4$ km s$^{-1}$. The emission concentrated near the disk's tidal truncation radius at $R_t = 56$ $R_\odot$ [25] (dashed line in the Doppler maps) is as follows:

$$\frac{R_t}{a} = \frac{0.6}{1 + q} \tag{1}$$

where $a$ is the distance between the system components, and $q = M_2/M_1$ is their mass ratio.

In the 2017−2019 map, the maximum intensity from the spot moved to $V_y \approx 100$ km s$^{-1}$ and $V_x \approx -13$ km s$^{-1}$, and another, less intense compact emission appeared on the opposite side of the disk at $V_y \approx$ -100 km s$^{-1}$ and $V_x \approx 32$ km s$^{-1}$. The Doppler map for the last time interval (2019–2021), when the disk emission was the weakest, has a similar structure with a small clockwise displacement of the spots. The brighter spot also looks more intense in comparison with the virtually invisible opposite one.

We measured the positions of the maximum intensity of the bright spot by fitting it to a single Gaussian in the X and Y direction on the Doppler maps. The measurement results are given in Table 2 and shown by a white line over the trailed spectra in Figure 3. The errors correspond to $1\sigma$ of the Gaussian fit and reflect the velocity dispersion within the spot. Assuming Keplerian velocities in the disk and that the spot is located in the disk's outer part, we estimate its radius $R_d$ using the following equation:

$$R_d = \frac{GM_1}{V_{spot}^2} \tag{2}$$

where $V_{spot}$ is the velocity at the maximum brightness of the hot spot on Doppler maps, $G$ is the gravitational constant, and $M$ is the mass of the star. The outer disk radii for the

different epochs are given in the last column of Table 2. As one can see in Figure 4, the disk radius decreases along with a decrease in $H_\alpha$ EW.

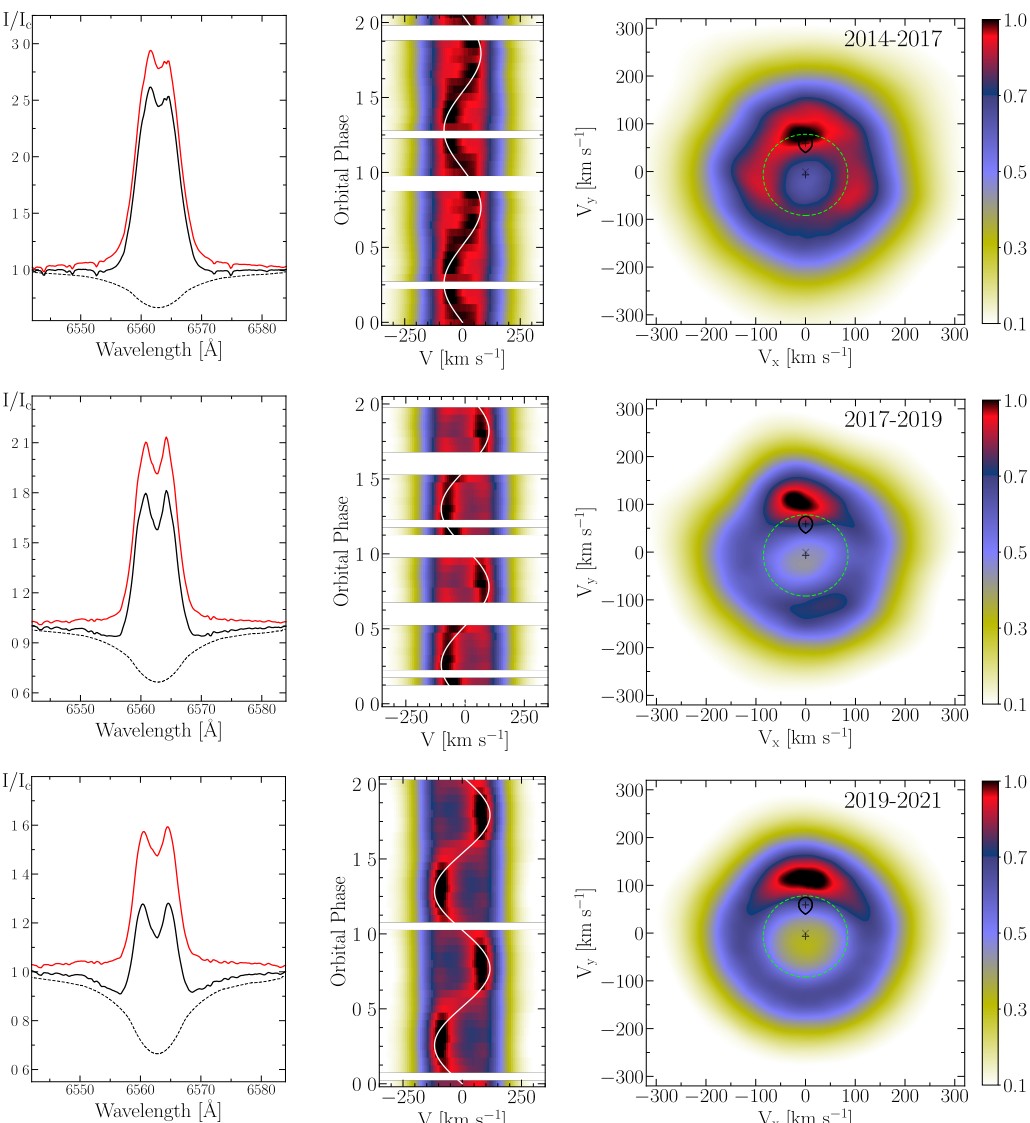

**Figure 3.** (**Left panels**): Average H$\alpha$ line profiles. Black solid line shows the original profile, dashed line is a model spectrum of the primary component's atmosphere, and solid red line is the result of subtraction of the model profile from the original one. (**Middle panels**): Reconstructed trailed spectra of the H$\alpha$ line folded with the orbital period P = 61.55 days. (**Right panels**): Doppler maps of the system. The map is centered at the system's center of mass, which is marked by the cross. The plus sign marks the center of mass of the primary component at $V_y \approx -7$ km s$^{-1}$ and $V_x = 0$ km s$^{-1}$. The center of mass of the secondary component is marked with another plus sign at $V_y \approx 50$ km s$^{-1}$ and $V_x = 0$ km s$^{-1}$ with the Roche lobe plotted around it. The dashed line marks $v \sin i = 85$ km s$^{-1}$ that corresponds to the tidal truncation disk radius. The color of the Doppler maps corresponds to arbitrary units of emission intensity (the yellow–blue–red–black palette corresponds to a change from low to high intensity).

Assuming Keplerian velocities of particles in the disk, we transformed the Doppler (velocity) map into the XY plane of the system (Figure 4). The color map in the XY plane clearly shows a strong extended emission from the disk's outer region, located on the line that connects the centers of the system components. The less massive companion cannot fill its Roche lobe; therefore, the excess of emissions here is probably due to tidal disturbances

of the disk by the companion, which provide a high velocity dispersion and deviations from Keplerian motion. The bright region on the opposite side of the secondary component is probably caused by a non-Keplerian motion in the disk, which is expected here. We believe that the bright spot is the source of the Hα line V/R variations.

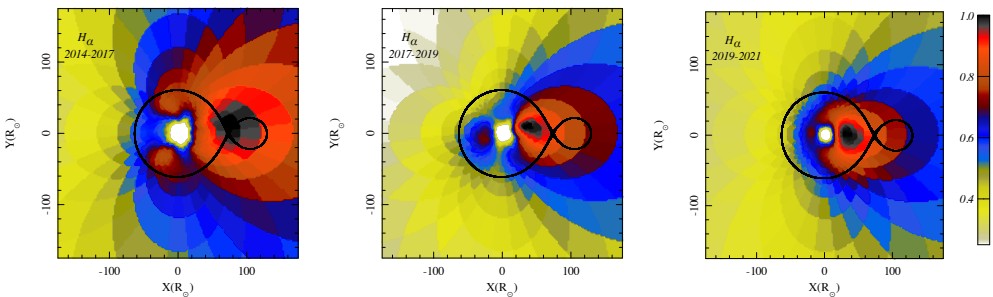

**Figure 4.** The brightness distribution in the disk, transformed from the Doppler map of the Hα emission line to the XY plane of the system. The colorbar shows normalized relative intensity. The Be star is located at the origin.

## 4. Discussion

Currently, κ Dra remains the only confirmed Be+sdB binary amidst a growing number of Be+sdO systems (e.g., [26]). In this research, we explored how the emitting region structure in the circumstellar disk κ Dra changed throughout the disk disappearance process. This tendency does not seem to be dependant on the secondary component's parameters. For instance, such a phenomenon was observed in π Aqr [21], where the secondary companion is either a main sequence star [27] or a white dwarf [28].

Despite the gradual decrease in the Hα line intensity, its V/R variations in the line profile remained phase-locked with the orbital period until the time when measurements could not be taken with a sufficient accuracy. This implies that as the density of the disk decreases, the distribution of the emitting matter in it remains stable. The Doppler maps constructed for different epochs of the κ Dra disk evolution show an emitting region produced by a density enhancement centred in the outer parts of the disk that faces the secondary companion. There is a number of Be + sdO binaries where the V/R variations are also locked with the orbital period (see [15] for a recent discussion) and may be caused by a similar mechanism.

Porter and Rivinius [29] described several models that were proposed to explain the circumstellar disk properties in Be stars. Theoretical explanations for the phase-locked V/R variations focus on the one-armed density waves in a near-Keplerian disk [5], which can generally reproduce the line profiles observed here [30]. Panoglou et al. [6] proposed that the phase-locked variations can be associated with a two-armed spiral structure in the disk, while longer-term V/R variations were considered to be caused by a one-armed spiral [31]. Tidal disturbances and disk heating from the side of the secondary companion are also possible contributors to the observed effects.

Based on the Doppler maps of κ Dra (see Figure 3), we suggest that the V/R variations are caused by a large hot spot located near the outer radius of the disk closest to the secondary companion. The spot position is stable as the disk radius decreases. The disk itself appears ragged, and the structure and location of the regions with various brightness levels slightly vary during disk evolution (see Figure 4). We suggest that the origin of this spot is mainly caused by irradiation and tidal effects induced on the disk by the secondary companion.

## 5. Conclusions

We explored the evolution of the region responsible for the Hα line emission in the κ Dra binary system throughout the disappearance process of the circumstellar disk around the primary component. The following results were obtained:

1.  Despite the gradual decrease in the Hα line intensity, its V/R variations remained phase-locked with the orbital period.
2.  The Doppler maps constructed throughout the disk evolution show the presence of a nearly stable bright emitting region located in the outer part of the disk in the direction of the secondary companion. We suggest that this structure is responsible for the V/R variations observed in the Hα line profile.
3.  The origin of the hot spot is most likely related to the formation of a one-armed spiral structure in the disk, tidal effects, and irradiation from the secondary companion.

The alternative modelling of the disk structure at various levels of emission-line strengths would be important to verify this hypothesis.

**Author Contributions:** Observations, A.S.M.; data reduction, A.S.M., A.A.; data analysis, I.A.G., A.A., A.S.M., and S.V.Z.; software A.S.M. and S.V.Z.; writing—original draft preparation, I.A.G.; writing—review and editing, A.S.M., S.V.Z., and S.A.K. All authors have read and agreed to the published version of the manuscript.

**Funding:** This research was funded by the Science Committee of the Ministry of Science and Higher Education of the Republic of Kazakhstan (Grant No. AP14972742).

**Data Availability Statement:** Original spectra reported in this study are available on request from A.S.M. via email at a_mirosh@uncg.edu.

**Acknowledgments:** This research made use of the SIMBAD database, operating at CDS, Strasbourg, France, and the SAO/NASA ADS and BeSS database, operating at LESIA, Observatoire de Meudon, France: http://basebe.obspm.fr (accessed on 2 March 2024). S.V.Z. acknowledges PAPIIT grant IN119323.

**Conflicts of Interest:** The authors declare no conflicts of interest.

## Abbreviations

The following abbreviations are used in this manuscript: RV—radial velocity, V/R—violet-to-red peak intensity ratio in a double-peaked emission-line profile, R—spectral resolving power, EW—equivalent width, TCO—Three College Observatory.

## Note

1   http://basebe.obspm.fr/basebe/ (accessed on 2 March 2024).

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
