# Peer review of "Doppler Tomography of the Circumstellar Disk of the Be Star κ Draconis"

_galaxies, doi:10.3390/galaxies12030023_

Round 1

Reviewer 1 Report

Comments and Suggestions for Authors

The paper presents optical spectroscopy of the binary classical Be star kappa Draconis. From these data the authors verify the physical parameters of the system and, by means of Doppler tomography techniques, they elaborate maps of the density distribution of the H alpha emitting material in the circumstellar disk, and its variation with time. The obtained results are valuable and of high scientific interest, as they contribute to the knowledge of the mechanism of disk formation and evolution in Be stars, which are not yet completely understood.

The paper is well written, the techniques used are well described and the results are adequately elaborated and convincingly exposed. The tables and figures are illustrative and properly laid out.

From these considerations I recommend the manuscript to be published in its present form.

Reviewer 2 Report

Comments and Suggestions for Authors

The reviewed paper is devoted to the analysis of the well known and unique Be+sdB binary system k Dra.
The authors performed Doppler tomography of the circumstellar disc, determined its parameters, and explained the periodic variability of its spectral features.
The process is described in the paper in sufficient detail, and the results obtained are unquestionable.
The qualification and experience of the team in Doppler tomography in general and in the study of this star in particular leaves no doubt about the necessity and expediency of the publication of the paper in Galaxies.
The only thing that could be recommended is to remind the reader of the value of the inclination angle in the caption to Table 2 (this value of i=50.0 is given on page 3).

Reviewer 3 Report

Comments and Suggestions for Authors

Dear authors,

I read your manuscript titled "Doppler tomography of the circumstellar disk of the Be star κ Draconi" with great interest. I think the manuscript is well-written, mostly clear, and present an interesting analysis of a well-known and long-studied Be binary.

I have one major comment, especially thinking of a non-expert audience: please explain better how the method of Doppler Tomography works. As this is the central part of the paper, it should be explained in detail which assumptions go in the method, which data is used exactly, which results from the interpretation of the maps, and the caveats that are associated with the method. Additionally, Fig. 3 indicates that there is emission coming from inside the disk truncation radius. Please discuss the interpretation and implications of this in the text.

Apart from the major comment mentioned above, I have a few minor comments listed below, which are sorted by order of occurrence:

- Fig.1: it remains unclear which spectra are shown, and what they should demonstrate. It looks like they follow a certain phase, but the ordering is unclear. Please either change this, or make clear which epochs/phases were chosen.
- page 3: please double check the number of spectra used in the analysis. It remains unclear whether 101 spectra were used for the analysis, or 131 as stated on page ...
- page 3, beginning of Sect. 3: in my opinion, Fig. 1 does not show a "strong sinusoidal character that is phase-locked with the orbital period". That only becomes apparent from Fig. 2.
- page 3: please clarify how the RVs are measured from the absorption lines (Line-profile fitting? CCF? Bisector?, ...)
- page 3: it remains unclear from the text which values that go into the mass function were derived from the data in this work, and which ones were taken from previous work. One example is the inclination, was it derived from the spectra, or was the one derived by Klement+ adopted?
- page 3: you mention "S-wave, which is typically associated with a hot spot in the disk" - I think this statement requires a citation.
- Table 1: you mention "Adopted parameters" which leads to additional confusion. Which of those are adopted, and which of those are derived in this work?
- Fig. 2: the color map chose here might not be ideal, as the years 2014 and 2021 (in red an pink) appear very similar and are thus hard to discern. Please consider changing the colormap. Also, it would be useful to indicate the intervals used for the further analysis in the bottom panel of Fig. 2.
- Table 2: vsini here corresponds to the radial velocity, while usually this is used for the rotational velocity of a star. To avoid confusions, please use a different nomenclature.
- Fig. 3: could you also indicate the Roche Lobe for the primary in the Doppler maps?
- Fig. 4 seems to indicate, that the secondary is engulfed in the disk, and that it does not truncate the disk as such, but rather leads to a hot spot or overdensity. Is this interpretation correct, and what is the implication of this?

Comments on the Quality of English Language

The use of the English is of high-quality. There are a few minor typos that will surely be smoothed out in the language editing process.
